# Tiny Security Hole: First-Order Vulnerability of Masked SEED and Its Countermeasure

**DOI:** 10.3390/s24185894

**Published:** 2024-09-11

**Authors:** Ju-Hwan Kim, Dong-Guk Han

**Affiliations:** Department of Financial Information Security, Kookmin University, Seoul 02707, Republic of Korea; zzzz2605@kookmin.ac.kr

**Keywords:** side-channel analysis, correlation power analysis, countermeasure, mask, SEED

## Abstract

Side-channel analysis is a type of cryptanalysis that utilizes the physical leakage of a cryptographic device. An adversary exploits the relationship between a physical leakage and the secret intermediate value of an encryption algorithm. In order to prevent side-channel analysis, the masking method was proposed. Several masking methods of the ISO/IEC 18033-3 standard encryption algorithm SEED have been proposed, as the Korean financial IC (integrated circuit) card standard (CFIP.ST.FINIC-01-2021) mandates using a robust implementation of SEED as an encryption algorithm against side-channel analyses. However, vulnerabilities were reported, except for with only one masking method. This study proposes the first-order vulnerability of that masking method. That is, an adversary is able to perform a side-channel analysis with the same complexity as an unprotected implementation. In order to fix this vulnerability, we revise the masking method with negligible additional overhead. Its vulnerability and security are theoretically verified and experimentally demonstrated. The round key of the existing masking method is revealed with only 210 power consumption traces, while that of the proposed masking method is not disclosed with 10,000 traces.

## 1. Introduction

The physical leakages of computing devices, such as power consumption or electromagnetic emissions, are related to the data being processed. Hence, a cryptographic device could leak secret information through physical leakage. Side-channel analysis is a type of cryptanalysis that exploits a relationship between a physical leakage and the secret intermediate value of an encryption algorithm [1,2,3,4,5]. For instance, a correlation power analysis (CPA) reveals the secret key by investigating the correlation between power consumption and the hypothetical intermediate value calculated with a guessed key and known information (e.g., plaintext or ciphertext) [6]. Therefore, a security designer should implement an encryption algorithm so that the physical leakage is independent of a secret intermediate value [7,8,9]. Masking is a countermeasure against CPA that splits secret intermediate values using random masks [10]. The cryptographic device computes each share of the intermediate value separately. Hence, an adversary is unable to directly evaluate each guessed key based on the observed power consumption.

SEED is the ISO/IEC 18033-3 standard encryption algorithm developed by the Korea Internet and Security Agency (KISA) for financial applications [11]. Specifically, the Korean financial IC (integrated circuit) card standard mandates the use of a side-channel analysis-resistant SEED as its encryption algorithm [12]. Hence, several SEED masking methods have been proposed. Chang Kyun Kim et al. proposed the first masking method [13]. It was designed for memory efficiency by implementing two S-Boxes with one masked inverse table. To the best of our knowledge, the first-order vulnerability, which is that an intermediate value related to a secret key that is not concealed with a mask, of that masking method has not been reported. HeeSeok Kim et al. enhanced its computational efficiency by modifying the masked S-Box table to reduce the conversion from Boolean masking to arithmetic masking [14]. However, HeeSeok Kim’s masking method could be vulnerable because an adversary is able to reveal the mask by investigating the shape of the side-channel information emitted when calculating carry tables [15]. Therefore, this study inspects the vulnerability of Chang Kyun Kim’s masking method. To the best of our knowledge, the only known attack on a masking method is [16]. They proposed an efficient second-order CPA [17] attack when shuffling [10] and masking countermeasures are implemented.

Most Boolean masking schemes struggle to reduce the overhead related to the substitution operation. Linear operations require a few additional XOR operations because they generally consist of linear operations. In contrast, the substitution operation requires a precomputed table related to the mask. Consequently, the security designer typically focuses on constructing a masking scheme related to the substitution operation; security evaluators also target the substitution operation’s output. Our motivation is that the intermediate value of the linear operation could be exploited to disclose secret information with much lower attack complexity. This security hole could occur while combining substitution layer outputs in linear operations [18].


**Our contributions**
Revealing the first-order vulnerability of the masked SEED:We propose the first-order vulnerability of the existing SEED masking method, the only method for which a vulnerability has not been reported. This vulnerability allows an adversary to reveal the secret key of the algorithm with the same difficulty as the unprotected SEED.Presenting a secure first-order SEED masking method:We patch the existing masking method to conceal every sensitive intermediate value. The proposed masking method does not use an additional random mask which necessitates a more expensive operation, such as a deterministic random number generator.Demonstrating the vulnerability of the existing countermeasure and the robustness of the proposed countermeasure:The methods’ vulnerability and robustness are demonstrated by performing a CPA and test vector leakage assessment (TVLA) [19]. The round keys of the existing masking method are revealed with only 210 traces, whereas the CPA of the proposed method failed to reveal the key with 10,000 traces.


## 2. Preliminaries

### 2.1. SEED

SEED is a 128-bit symmetric key encryption algorithm with a 16-round Feistel structure [11]. Figure 1 shows the structure of its Feistel function *F*. The function first computes the XOR (⊕) between the 32-bit inputs (*C* and *D*) and round keys ($K_{i,0}$ and $K_{i,1}$). Next, it calculates function *G* and the 32-bit modular addition (⊞) three times. Function *G* consists of two 8-bit S-Boxes $S_{1}$, $S_{2}$ permutated by ANDing with the constants $m_{0}=0x\mathrm{fc},\;m_{1}=0\mathrm{xf}3,\;m_{2}=0\mathrm{xcf}$, and $m_{3}=0x3f$ and XORing four bytes. That is, the permutation is composed of a 6-bit extraction of each of the four S-Box outputs and their XORing. The SEED S-Boxes are defined as follows:(1)$$\begin{array}{c} \begin{array}{cc} S_{1}\left(x\right) & =A_{1}\cdot x^{247}⊕0\mathrm{xa}9,\\ S_{2}\left(x\right) & =A_{2}\cdot x^{251}⊕0x38, \end{array} \end{array}$$
where $A_{1}$ and $A_{2}$ are constant binary matrices and *x* is an element of the finite field $Z_{2^{8}}$.

### 2.2. CPA and TVLA

CPA is a traditional side-channel analysis method that exploits the relationship between the power consumption of a cryptographic device and the intermediate value related to its secret key. An adversary first chooses a proper power model that describes the relationship between the manipulated data and the device’s power consumption. For instance, an adversary may use the Hamming weight model, which assumes that the power consumption is linearly related to the number of bit 1s of data. The power model allows the adversary to estimate the power consumption corresponding to the intermediate value. Thus, the adversary is able to evaluate each candidate key in the keyspace. The adversary calculates the hypothetical intermediate value by assuming that the secret key is the candidate key. Then, the candidate key is evaluated in terms of the linearity between the power consumption and the hypothetical power consumption, which is calculated by inputting the hypothetical intermediate value into the power model. The linearity is measured by the Pearson correlation coefficient. If the candidate key is correct, then the hypothetical power consumption is linearly related to the power consumption; the absolute correlation coefficient is close to 1. If the candidate key is incorrect, then the hypothetical intermediate value is mostly not equal to the actual intermediate value; the correlation coefficient is close to 0. The adversary decides that the candidate key with the highest absolute correlation coefficient is the secret key. In order to evaluate the performance of a CPA, the guessing entropy and the success rate are typically employed [20]. The guessing entropy is defined as the average rank of the secret key when sorting keys by their corresponding correlation coefficients. The success rate is the proportion of attacks where the secret key is sorted first among several attacks. The convergence of the guessing entropy and success rate to 1 indicates that every CPA successfully discloses the secret key.

In order to evaluate CPA resistance, TVLA is generally utilized [19]. TVLA measures whether power consumption depends on a secret intermediate value. An evaluator first divides the power consumption traces into two sets using a sensitive intermediate value. For instance, one set is constructed using traces of specific fixed plaintext, while another is random plaintext traces. Next, the statistical difference between the two sets is measured by Welch’s *t*-test, a statistical hypothesis test. The null hypothesis of the *t*-test is that the two population means are equal. Thus, rejecting the null hypothesis suggests that the side-channel trace depends on the secret intermediate value. The statistic T-value is defined as follows:$$T=\frac{\mu_{A}-\mu_{B}}{\frac{\sigma_{A}^{2}}{N_{A}}+\frac{\sigma_{B}^{2}}{N_{B}}},$$
where $N_{A}$, $\mu_{A}$, and $\sigma_{A}$ are the size, mean, and standard deviation of *A*, respectively. If the absolute T-value exceeds the threshold, the null hypothesis is rejected in favor of the alternative hypothesis. For instance, if the absolute T-value is greater than 4.5, the null hypothesis is rejected with a confidence level of 99.999%.

### 2.3. Existing SEED Masking Method

Chang Kyun Kim et al. proposed a memory-efficient masking method that implements two S-Boxes with one masked lookup table $M^{-1}$, as illustrated in Figure 2 [11].

The power of *x* in Equation (Equation 1) can be calculated by $x^{247}=x^{255-8}={\left(x^{-1}\right)}^{8},\;x^{251}=x^{255-4}={\left(x^{-1}\right)}^{4}$ because *x* is an element of the finite field $Z_{2^{8}}$. Therefore, for constant binary matrices $C_{1}$ and $C_{2}$, the S-Boxes are expressed as follows:$$\begin{array}{cccc} S_{1}\left(x\right) & =A_{1}\cdot x^{247}⊕0\mathrm{xa}9 & = & C_{1}\cdot x^{-1}⊕0\mathrm{xa}9,\\ S_{2}\left(x\right) & =A_{2}\cdot x^{251}⊕0x38 & = & C_{2}\cdot x^{-1}⊕0x38. \end{array}$$

The above equations signify that only the lookup table for the masked inverse operation is required to implement two S-Boxes. The masked inverse lookup table $M^{-1}$ is defined as follows:$$M^{-1}\left[x⊕M_{1}\right]=x^{-1}⊕M_{4},$$
where $M_{1}$ and $M_{4}$ are random 8-bit numbers. As shown in Figure 2, masked the S-Boxes are calculated as follows:$$\begin{array}{cccc} MaskedS_{1}\left(x⊕M_{1}\right) & =C_{1}\cdot M^{-1}\left[x⊕M_{1}\right]⊕M_{6} & = & C_{1}\cdot \left(x^{-1}⊕M_{4}\right)⊕M_{6}\\ & =C_{1}\cdot x^{-1}⊕C_{1}\cdot M_{4}⊕M_{6} & = & C_{1}\cdot x^{-1}⊕M_{5}⊕0\mathrm{xa}9⊕M_{6}\\ & =S_{1}\left(x\right)⊕M_{3}, & & \\ MaskedS_{2}\left(x⊕M_{1}\right) & =C_{2}\cdot M^{-1}\left[x⊕M_{1}\right]⊕M_{1} & = & C_{2}\cdot \left(x^{-1}⊕M_{4}\right)⊕M_{1}\\ & =C_{2}\cdot x^{-1}⊕C_{2}\cdot M_{4}⊕M_{1} & = & C_{2}\cdot x^{-1}⊕M_{2}⊕0x38⊕M_{1}\\ & =S_{2}\left(x\right)⊕M_{3}, & \end{array}$$
where $M_{2},M_{3},M_{5}$, and $M_{6}$ are driven masks from $M_{1}$ and $M_{4}$ and defined as follows:$$\begin{array}{cccc} M_{2} & =C_{2}\cdot M_{4}⊕0x38,\;M_{3} & = & M_{1}⊕M_{2},\\ M_{5} & =C_{1}\cdot M_{4}⊕0\mathrm{xa}9,\;M_{6} & = & M_{3}⊕M_{5}. \end{array}$$

## 3. First-Order Vulnerability of Masked SEED

### 3.1. Theoretical Analysis

The motivation of the proposed method is that the operands of the last XOR operation of the function *G* are concealed with the same mask, $M_{3}$. As XORing two masked intermediate values induces a canceling of the mask, the adversary is able to exploit the XOR output as an intermediate value of the first-order CPA. Even though the operands are ANDed before the XOR operation, a part of the mask is still canceled. The SEED AND constants $m_{0}$, $m_{1}$, $m_{2}$, and $m_{3}$ are designed to extract 6 bits of the S-Box output. Each $m_{i}$ is the left rotation of $m_{i-1}$ by 2. The binary representation of the constants is presented as follows:$$\begin{array}{cccc} m_{0} & ={\left(1111\;1100\right)}_{2},\;m_{1} & = & {\left(1111\;0011\right)}_{2},\\ m_{2} & ={\left(1100\;1111\right)}_{2},\;m_{3} & = & {\left(0011\;1111\right)}_{2}. \end{array}$$

Let us define $m_{i\&j}=m_{i}\wedge m_{j}$ and $m_{i-j}=m_{i}\wedge \bar{m_{j}}$; the bit of $m_{i-j}$ is set as one if, and only if, the corresponding bit of $m_{i}$ is one but the bit of $m_{j}$ is zero. That is, $m_{i-j}\wedge m_{i\&j}=0$ and $m_{i-j}⊕m_{i\&j}=m_{i}$. Similarly, let us define $m_{i\&j\&k}=m_{i}\wedge m_{j}\wedge m_{k}$ and $m_{i\&j-k}=m_{i\&j}\wedge \bar{m_{k}}$. The definition of the AND constants directly drives the two properties below:
**Property** **1.***For all distinct $i,\;j\in \left\{0,1,2,3\right\},\mathrm{HW}\left(m_{i\&j}\right)=4$ and $\mathrm{HW}\left(m_{i-j}\right)=2$.*
**Property** **2.***For all distinct $i,\;j,\;k\in \left\{0,1,2,3\right\},m_{i\&j-k}=m_{i-k}=m_{j-k}$.*
HW denotes the Hamming weight, the number of 1s in binary representation.

XOR is a binary operator; the last XOR does not calculate four operands simultaneously. In order to investigate intermediate values during the last XOR, we utilize the following property:
**Property** **3.***$\left(A\wedge C\right)⊕\left(B\wedge C\right)=\left(A⊕B\right)\wedge C$.*
**Proof.** The XOR between *A* and *B* can be implemented by AND (∧) and OR (∨) as $\left(A\wedge \bar{B}\right)\vee \left(\bar{A}\wedge B\right)$, where $\bar{A}$ is NOT *A*. Then, the following is derived:
$$\begin{array}{cc} \; & \left(A⊕B\right)\wedge C\\ \; & =\left(\left(A\wedge \bar{B}\right)\vee \left(\bar{A}\wedge B\right)\right)\wedge C\\ \; & =\left(A\wedge \bar{B}\wedge C\right)\vee \left(\bar{A}\wedge B\wedge C\right)\\ \; & =\left(A\wedge \bar{B}\wedge C\right)\vee \left(\bar{A}\wedge B\wedge C\right)\vee 0\\ \; & =\left(A\wedge \bar{B}\wedge C\right)\vee \left(\bar{A}\wedge B\wedge C\right)\vee \left(\left(A\vee B\right)\wedge 0\right)\\ \; & =\left(A\wedge \bar{B}\wedge C\right)\vee \left(\bar{A}\wedge B\wedge C\right)\vee \left(\left(A\vee B\right)\wedge \left(\bar{C}\wedge C\right)\right)\\ \; & =\left(A\wedge \bar{B}\wedge C\right)\vee \left(\bar{A}\wedge B\wedge C\right)\vee \left(A\wedge \bar{C}\wedge C\right)\vee \left(B\wedge \bar{C}\wedge C\right)\\ \; & =\left(A\wedge C\wedge \left(\bar{B}\vee \bar{C}\right)\right)\vee \left(\left(\bar{A}\vee \bar{C}\right)\wedge B\wedge C\right)\\ \; & =\left(\left(A\wedge C\right)\wedge \bar{\left(B\wedge C\right)}\right)\vee \left(\bar{\left(A\wedge C\right)}\wedge \left(B\wedge C\right)\right)\\ \; & =\left(A\wedge C\right)⊕\left(B\wedge C\right). \end{array}$$ □

Without a loss of generality, let the XOR calculate four operands in order, from the left to the right of Figure 2. The first operands are $\left(S_{2}\left[X_{3}\right]⊕M_{3}\right)\wedge m_{2}$ and $\left(S_{1}\left[X_{2}\right]⊕M_{3}\right)\wedge m_{1}$. Property 3 enables splitting the output of the first XOR, as shown below:(2)$$\begin{array}{c} \begin{array}{cc} \; & \left(\left(S_{2}\left[X_{3}\right]⊕M_{3}\right)\wedge m_{2}\right)⊕\left(\left(S_{1}\left[X_{2}\right]⊕M_{3}\right)\wedge m_{1}\right)\\ = & \left(\left(S_{2}\left[X_{3}\right]⊕M_{3}\right)\wedge \left(m_{1\&2}⊕m_{2-1}\right)\right)⊕\left(\left(S_{1}\left[X_{2}\right]⊕M_{3}\right)\wedge \left(m_{1\&2}⊕m_{1-2}\right)\right)\\ = & \left(\left(S_{2}\left[X_{3}\right]⊕M_{3}\right)\wedge m_{1\&2}\right)⊕\left(\left(S_{2}\left[X_{3}\right]⊕M_{3}\right)\wedge m_{2-1}\right)⊕\\ \; & \left(\left(S_{1}\left[X_{2}\right]⊕M_{3}\right)\wedge m_{1\&2}\right)⊕\left(\left(S_{1}\left[X_{2}\right]⊕M_{3}\right)\wedge m_{1-2}\right)\\ = & \left(\left(S_{2}\left[X_{3}\right]⊕S_{1}\left[X_{2}\right]\right)\wedge m_{1\&2}\right)⊕\left(\left(S_{2}\left[X_{3}\right]⊕M_{3}\right)\wedge m_{2-1}\right)⊕\left(\left(S_{1}\left[X_{2}\right]⊕M_{3}\right)\wedge m_{1-2}\right). \end{array} \end{array}$$
As the output of AND between any two elements in the set $\left\{m_{1\&2},\;m_{2-1},\;m_{1-2}\right\}$ is zero, Equation (Equation 2) drives the following:$$\begin{array}{cc} \left(\left(\left(S_{2}\left[X_{3}\right]⊕M_{3}\right)\wedge m_{2}\right)⊕\left(\left(S_{1}\left[X_{2}\right]⊕M_{3}\right)\wedge m_{1}\right)\right)\wedge m_{1\&2} & =\left(S_{2}\left[X_{3}\right]⊕S_{1}\left[X_{2}\right]\right)\wedge m_{1\&2},\\ \left(\left(\left(S_{2}\left[X_{3}\right]⊕M_{3}\right)\wedge m_{2}\right)⊕\left(\left(S_{1}\left[X_{2}\right]⊕M_{3}\right)\wedge m_{1}\right)\right)\wedge m_{2-1} & =\left(S_{2}\left[X_{3}\right]⊕M_{3}\right)\wedge m_{2-1},\\ \left(\left(\left(S_{2}\left[X_{3}\right]⊕M_{3}\right)\wedge m_{2}\right)⊕\left(\left(S_{1}\left[X_{2}\right]⊕M_{3}\right)\wedge m_{1}\right)\right)\wedge m_{1-2} & =\left(S_{1}\left[X_{2}\right]⊕M_{3}\right)\wedge m_{1-2}. \end{array}$$
The above equations indicate that the first XOR output consists of four bits of the unmasked XORed S-Box outputs $S_{2}\left[X_{3}\right]⊕S_{1}\left[X_{2}\right]$ and two bits of the masked $S_{2}\left[X_{3}\right]⊕M_{3}$ and $S_{1}\left[X_{2}\right]⊕M_{3}$ attributed to Property 1.

Likewise, the output of the second XOR, whose operand is the first XOR output and $S_{2}\left[X_{1}\right]⊕M_{3}$, consists of six bits of the unmasked intermediate value. As the output of AND between any two elements in $\{m_{0\&1\&2},\;m_{0\&1-2},\;m_{1\&2-0},$$m_{0\&2-1}\}$ is zero, the second XOR output can be split as shown in Equation (Equation 3). The second XOR output consists of two bits of each of the unmasked XORed S-Box outputs $S_{2}\left[X_{3}\right]⊕S_{1}\left[X_{2}\right]$, $S_{2}\left[X_{3}\right]⊕S_{2}\left[X_{1}\right]$, and $S_{1}\left[X_{2}\right]⊕S_{2}\left[X_{1}\right]$ and two bits of masked $S_{2}\left[X_{3}\right]⊕S_{1}\left[X_{2}\right]⊕S_{2}\left[X_{1}\right]⊕M_{3}$.
(3)$$\begin{array}{c} \begin{array}{cc} \; & \left(\left(S_{2}\left[X_{3}\right]⊕S_{1}\left[X_{2}\right]\right)\wedge m_{1\&2}\right)⊕\left(\left(S_{2}\left[X_{3}\right]⊕M_{3}\right)\wedge m_{2-1}\right)⊕\left(\left(S_{1}\left[X_{2}\right]⊕M_{3}\right)\wedge m_{1-2}\right)\\ \; & ⊕\left(\left(S_{2}\left[X_{1}\right]⊕M_{3}\right)\wedge m_{0}\right)\\ = & \left(\left(S_{2}\left[X_{3}\right]⊕S_{1}\left[X_{2}\right]\right)\wedge \left(m_{1\&2-0}⊕m_{0\&1\&2}\right)\right)\\ \; & ⊕\left(\left(S_{2}\left[X_{3}\right]⊕M_{3}\right)\wedge m_{2-1}\right)⊕\left(\left(S_{1}\left[X_{2}\right]⊕M_{3}\right)\wedge m_{1-2}\right)⊕\left(\left(S_{2}\left[X_{1}\right]⊕M_{3}\right)\wedge m_{0}\right)\\ = & \left(\left(S_{2}\left[X_{3}\right]⊕S_{1}\left[X_{2}\right]\right)\wedge m_{1\&2-0}\right)⊕\left(\left(S_{2}\left[X_{3}\right]⊕S_{1}\left[X_{2}\right]\right)\wedge m_{0\&1\&2}\right)\\ \; & ⊕\left(\left(S_{2}\left[X_{3}\right]⊕M_{3}\right)\wedge m_{2-1}\right)⊕\left(\left(S_{1}\left[X_{2}\right]⊕M_{3}\right)\wedge m_{1-2}\right)⊕\left(\left(S_{2}\left[X_{1}\right]⊕M_{3}\right)\wedge m_{0}\right)\\ = & \left(\left(S_{2}\left[X_{3}\right]⊕S_{1}\left[X_{2}\right]\right)\wedge m_{1-0}\right)⊕\left(\left(S_{2}\left[X_{3}\right]⊕S_{1}\left[X_{2}\right]\right)\wedge m_{0\&1\&2}\right)\\ \; & ⊕\left(\left(S_{2}\left[X_{3}\right]⊕M_{3}\right)\wedge m_{2-1}\right)⊕\left(\left(S_{1}\left[X_{2}\right]⊕M_{3}\right)\wedge m_{1-2}\right)⊕\left(\left(S_{2}\left[X_{1}\right]⊕M_{3}\right)\wedge m_{0}\right)\\ \; & \left(∵\mathrm{Property}\;\right)\\ = & \left(\left(S_{2}\left[X_{3}\right]⊕S_{1}\left[X_{2}\right]\right)\wedge m_{1-0}\right)⊕\left(\left(S_{2}\left[X_{3}\right]⊕S_{1}\left[X_{2}\right]\right)\wedge m_{0\&1\&2}\right)\\ \; & ⊕\left(\left(S_{2}\left[X_{3}\right]⊕M_{3}\right)\wedge m_{2-1}\right)⊕\left(\left(S_{1}\left[X_{2}\right]⊕M_{3}\right)\wedge m_{1-2}\right)\\ \; & ⊕\left(\left(S_{2}\left[X_{1}\right]⊕M_{3}\right)\wedge m_{2-1}\right)⊕\left(\left(S_{2}\left[X_{1}\right]⊕M_{3}\right)\wedge m_{1-2}\right)\\ \; & ⊕\left(\left(S_{2}\left[X_{1}\right]⊕M_{3}\right)\wedge m_{0\&1\&2}\right)\\ \; & \left(∵m_{0}=m_{0-1}⊕m_{0\&1}=m_{0-1}⊕m_{0\&1-2}⊕m_{0\&1\&2}=m_{2-1}⊕m_{1-2}⊕m_{0\&1\&2}\right)\\ = & \left(\left(S_{2}\left[X_{3}\right]⊕S_{1}\left[X_{2}\right]\right)\wedge m_{1-0}\right)\\ \; & ⊕\left(\left(S_{2}\left[X_{3}\right]⊕S_{1}\left[X_{2}\right]⊕S_{2}\left[X_{1}\right]⊕M_{3}\right)\wedge m_{0\&1\&2}\right)\\ \; & ⊕\left(\left(S_{2}\left[X_{3}\right]⊕S_{2}\left[X_{1}\right]\right)\wedge m_{2-1}\right)\\ \; & ⊕\left(\left(S_{1}\left[X_{2}\right]⊕S_{2}\left[X_{1}\right]\right)\wedge m_{1-2}\right). \end{array} \end{array}$$

Equations (Equation 2) and (Equation 3) evidently verify the presence of an unmasked secret intermediate value. Namely, the existing masking method has a first-order vulnerability and enables the adversary to perform a first-order CPA with the XORed S-Box outputs. If the adversary knows the order of the XORs, the adversary can utilize the exact intermediate value, such as $\left(S_{2}\left[X_{3}\right]⊕S_{1}\left[X_{2}\right]\right)\wedge m_{1\&2}$ in Equation (Equation 2). Although the order is unkown to the adversary, the adversary can still perform a first-order CPA with the 8-bit XORed S-Box outputs. The left bits, which are not part of the intermediate value, like $\left(S_{2}\left[X_{3}\right]⊕S_{1}\left[X_{2}\right]\right)\wedge \bar{m_{1\&2}}$ in Equation (Equation 2), drop the correlation. However, the 8-bit XORed S-Box outputs are still related to the power consumption.

The limitation of the proposed method is that it may reveal three of four bytes of the round key depending on the order of the XOR operation. Suppose that every four bytes of function *G*’s output is calculated by XORing four S-Box inputs in the same order. In this case, the round key associated with the last operand is unrelated to the proposed intermediate value, and the adversary reveals only three bytes of the round key; the round key related to the last operand is not revealed. On the other hand, if XOR calculates four operands in a different order, then the adversary reveals four bytes of the round key.

### 3.2. Experiments

The proposed method is demonstrated using the side-channel evaluation board ChipWhisperer-Lite XMEGA. A SEED compiled by avr-gcc 11.1.0 is executed on an 8-bit microcontroller ATXmega128D4-AU, and its power consumption during the calculation of the first function *G* is collected. The function *G* calculates the last XORs, from the left to the right of Figure 2. We measured 10,000 power consumption traces for each of the random plaintext and fixed plaintext encryptions for the TVLA. A CPA was performed ten times with different 10,000 traces to calculate the guessing entropy and success rate.

Figure 3 shows the point-wise T-value and the absolute correlation coefficient of the right intermediate value. The maximum absolute T-value of power consumption during the XOR operation is 76.1, which is considerably higher than the threshold of 4.5 (represented by the red line in the middle subplot of Figure 3). This outcome signifies that the null hypothesis is rejected with a confidence level of 99.999%. That is, the power consumption is related to the input; the mask does not conceal some of the intermediate values related to the input. For the CPA, we assume a practical attacker model in which the adversary does not know the order of the XORs; the 8 bits of the XORed S-Box outputs are used as an intermediate value. The bottom subplot of Figure 3 shows the absolute correlation coefficient of each intermediate value. $\left(X_{3},\;X_{2}\right)$ in the legend indicates that the intermediate value is $S_{2}\left[X_{3}\right]⊕S_{1}\left[X_{2}\right]$. As expected, in Section 3.1, the power consumption is related to the secret intermediate value. The correlation of the intermediate value derived from the front three XOR operands is significantly high. As an example, $\left(X_{3},\;X_{2}\right)$, $\left(X_{3},\;X_{0}\right)$, and $\left(X_{2},\;X_{1}\right)$ are the proposed intermediate values for the first XOR operation over four inputs. The correlations of those intermediate values are significantly high during the first XOR operation (from 980 to 1640 points).

Figure 4 shows the absolute correlation coefficient of the correct and other keys according to the number of traces performed. If the number of traces exceeds 210, then the correlation of the correct key is greater than that of the 65,535 other incorrect keys. The correct key is evidently distinguishable if the number of traces exceeds 300. Figure 5 shows a guessing entropy and success rate that converge to one if the number of traces exceeds 210. Each CPA successfully reveals each round key with 210 traces. The CPA results exhibit the presence of a first-order vulnerability in the existing masking scheme.

## 4. Proposed Masking Scheme

### 4.1. Masking Scheme

The existing masking method has a first-order vulnerability because its S-Box outputs are concealed with the same mask. We patch this method by concealing the S-Box outputs with different masks. Figure 6 illustrates the proposed masking method. The dissimilarity between the proposed masking method and the existing method is highlighted. The outputs of $C_{1}$ and $C_{2}$ are already concealed with different masks, $M_{2}$ and $M_{5}$, respectively. Therefore, the proposed masking method only remasks the first two bytes by XORing $M_{1}^{\prime }$ and $M_{2}^{\prime }$ instead of $M_{1}$ and $M_{6}$, respectively, where $M_{1}^{\prime }=C_{2}\cdot M_{1}$ and $M_{2}^{\prime }=C_{1}\cdot M_{1}$. After XORing four S-Box outputs, they are remasked to $M_{3}$ by XORing $M_{3}^{\prime }$, $M_{4}^{\prime }$, $M_{5}^{\prime }$, and $M_{6}^{\prime }$, where
$$\begin{array}{cc} M_{3}^{\prime }= & \left(\left(M_{1}⊕M_{1}^{\prime }\right)\wedge m_{2}\right)⊕\left(\left(M_{6}⊕M_{2}^{\prime }\right)\wedge m_{1}\right)⊕\left(M_{1}\wedge m_{0}\right)⊕\left(M_{6}\wedge m_{3}\right),\\ M_{4}^{\prime }= & \left(\left(M_{1}⊕M_{1}^{\prime }\right)\wedge m_{1}\right)⊕\left(\left(M_{6}⊕M_{2}^{\prime }\right)\wedge m_{0}\right)⊕\left(M_{1}\wedge m_{3}\right)⊕\left(M_{6}\wedge m_{2}\right),\\ M_{5}^{\prime }= & \left(\left(M_{1}⊕M_{1}^{\prime }\right)\wedge m_{0}\right)⊕\left(\left(M_{6}⊕M_{2}^{\prime }\right)\wedge m_{3}\right)⊕\left(M_{1}\wedge m_{2}\right)⊕\left(M_{6}\wedge m_{1}\right),\\ M_{6}^{\prime }= & \left(\left(M_{1}⊕M_{1}^{\prime }\right)\wedge m_{3}\right)⊕\left(\left(M_{6}⊕M_{2}^{\prime }\right)\wedge m_{2}\right)⊕\left(M_{1}\wedge m_{1}\right)⊕\left(M_{6}\wedge m_{0}\right). \end{array}$$

The proposed masking method does not use an additional random mask. Generating a random mask substantially increases the computational overhead because it requires more expensive operations, such as a deterministic random number generator. The proposed masking method requires only two more XOR operations per function *G*. The generation of masking requires a supplementary six bytes of memory, 20 XORs, and 16 ANDs. The additional overhead is negligible because the conversion from Boolean masking to arithmetic masking is the primary overhead of the masking method.

In order to validate the security of the proposed masking scheme, Property 4 is defined as follows:
**Property** **4.***$M_{2}⊕M_{5}$ and $M_{1}^{\prime }⊕M_{2}^{\prime }$ are uniformly distributed if $M_{4}$ and $M_{1}$ are uniformly chosen.*
**Proof.** The definitions of $C_{1}$ and $C_{2}$ drive $\left(C_{2}\cdot i\right)⊕\left(C_{1}\cdot i\right)\neq \left(C_{2}\cdot j\right)⊕\left(C_{1}\cdot j\right)$ for any distinct $i,\;j\in \left\{0,\;1,\;\cdots ,\;255\right\}$. □

Recall that $M_{2}=C_{2}\cdot M_{4}⊕0x38$, $M_{5}=C_{1}\cdot M_{4}⊕0\mathrm{xa}9$, $M_{1}^{\prime }=C_{2}\cdot M_{1}$, and $M_{2}^{\prime }=C_{1}\cdot M_{1}$. The outputs of XOR between the two masks concealing the S-Box output are as shown below:$$\begin{array}{cc} \left(M_{2}⊕M_{1}^{\prime }\right)⊕\left(M_{5}⊕M_{2}^{\prime }\right) & =\left(M_{2}⊕M_{5}\right)⊕\left(M_{1}^{\prime }⊕M_{2}^{\prime }\right),\\ \left(M_{2}⊕M_{1}^{\prime }\right)⊕M_{2} & =M_{1}^{\prime },\\ \left(M_{2}⊕M_{1}^{\prime }\right)⊕M_{5} & =\left(M_{2}⊕M_{5}\right)⊕M_{1}^{\prime },\\ \left(M_{5}⊕M_{2}^{\prime }\right)⊕M_{2} & =\left(M_{2}⊕M_{5}\right)⊕M_{2}^{\prime },\\ \left(M_{5}⊕M_{2}^{\prime }\right)⊕M_{5} & =M_{2}^{\prime },\\ M_{2}⊕M_{5} & =M_{2}⊕M_{5}. \end{array}$$

The above masks are uniformly distributed because of Property 4. For example, $\left(M_{2}⊕M_{5}\right)$ and $\left(M_{1}^{\prime }⊕M_{2}^{\prime }\right)$ are driven by independently chosen masks $M_{4}$ and $M_{1}$, respectively, and each is uniformly distributed. Therefore, the XOR output is also uniformly distributed.

The outputs of XOR between three masks concealing the S-Box output are shown below:$$\begin{array}{cc} \left(M_{5}⊕M_{2}^{\prime }\right)⊕M_{2}⊕M_{5} & =M_{2}⊕M_{2}^{\prime },\\ \left(M_{2}⊕M_{1}^{\prime }\right)⊕M_{2}⊕M_{5} & =M_{1}^{\prime }⊕M_{5},\\ \left(M_{2}⊕M_{1}^{\prime }\right)⊕\left(M_{5}⊕M_{2}^{\prime }\right)⊕M_{5} & =M_{2}⊕\left(M_{1}^{\prime }⊕M_{2}^{\prime }\right),\\ \left(M_{2}⊕M_{1}^{\prime }\right)⊕\left(M_{5}⊕M_{2}^{\prime }\right)⊕M_{2} & =M_{5}⊕\left(M_{1}^{\prime }⊕M_{2}^{\prime }\right). \end{array}$$
Similarly, $M_{2}$ and $M_{5}$ are derived from mask $M_{4}$, and $M_{1}^{\prime }$, $M_{2}^{\prime }$, and $M_{1}^{\prime }⊕M_{2}^{\prime }$ are derived from mask $M_{1}$. Because $M_{1}$ and $M_{4}$ are independent, the output of XOR between any three masks is uniform.

### 4.2. Experiments

TVLA and CPA were performed in the same environment as in Section 3. All T-values are less than 4.5, as shown in the middle subplot of Figure 7; the maximum T-value is only 3.9. This result suggests that the difference between the means of the random plaintext traces and the fixed plaintext traces is insignificant. The proposed masking method successfully conceals intermediate values related to the input. In terms of the CPA, while the maximum absolute correlation coefficient of the existing masking method is 0.64, as shown in Figure 3, that of the proposed masking method is only as large as 0.04. This result suggests that the power consumption is unrelated to the proposed intermediate value, as shown in the bottom subplot.

CPA fails to reveal the secret key of the proposed masking method, as shown in Figure 8 and Figure 9. The absolute correlation coefficient of the correct key is less than that of an incorrect key for each S-Box output combination, as shown in Figure 8. Recall that the round key of the existing masking method was revealed with only 210 traces, and the correct key was clearly distinguishable from the incorrect keys, as shown in Figure 4. The correlation of the correct key when using the proposed method is indistinguishable even with 10,000 traces, and increasing the number of traces does not increase the correlation coefficient of the correct key. Each CPA fails to reveal the secret key, as shown in Figure 9. The guessing entropy is around 33,000, which is the expectation when the adversary randomly chooses the key. Moreover, increasing the number of traces does not decrease the guessing entropy. The results suggest that the proposed masking method does not have a first-order vulnerability.

We measured the number of clock cycles for each SEED component to compare the overhead between the existing and proposed countermeasures, as shown in Table 1 The ratio in the table is calculated by dividing the number of clock cycles of the proposed masking method by that of the existing masking method. As discussed in Section 4.1, the increase in overhead is negligible. The conversion from Boolean masking to arithmetic masking primarily contributes to the masking overhead. Although the number of clock cycles for the function G increased by 17 percentage points, the overall increase in encryption, including masking generation, is only 0.75 percentage points.

## 5. Conclusions

The existing masking method of the SEED has a first-order vulnerability because its S-Box outputs are concealed with the same mask. The mask is inevitably canceled because the function *G* performs XOR with its S-Box outputs. Hence, a CPA adversary is able to utilize the intermediate value of the last XOR operation and perform a first-order CPA on the masked SEED. That is, the adversary is able to disclose the secret key with the same complexity required as for its unprotected implementation. In order to fix the security hole, we proposed a masking method that concealed each S-Box output with a different mask but required negligible overhead. Its vulnerability and security were theoretically verified and practically demonstrated. A CPA performed on the existing masking method discloses the correct key with only 210 traces, whereas a CPA performed on the proposed masking method failed even with 10,000 traces.

The proposed security hole could also be present in other encryption algorithms. Although the AND operation is insufficient to prevent mask canceling, a security designer might mistakenly design a masking scheme to conceal the XOR operand with the same mask, like SEED. For instance, the block cipher ARIA also performs XOR on four S-Box outputs after the AND operation. Consequently, some ARIA masking methods might also have the same security holes as SEED masking. Our future work will investigate another masking scheme that has the proposed vulnerability and patch it.

## Figures and Tables

**Figure 1 sensors-24-05894-f001:**
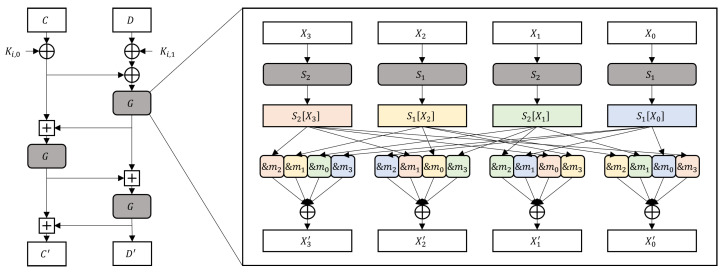
Structure of the SEED Feistel function *F*. A rounded rectangle is an operation, and a sharp rectangle is an intermediate value.

**Figure 2 sensors-24-05894-f002:**
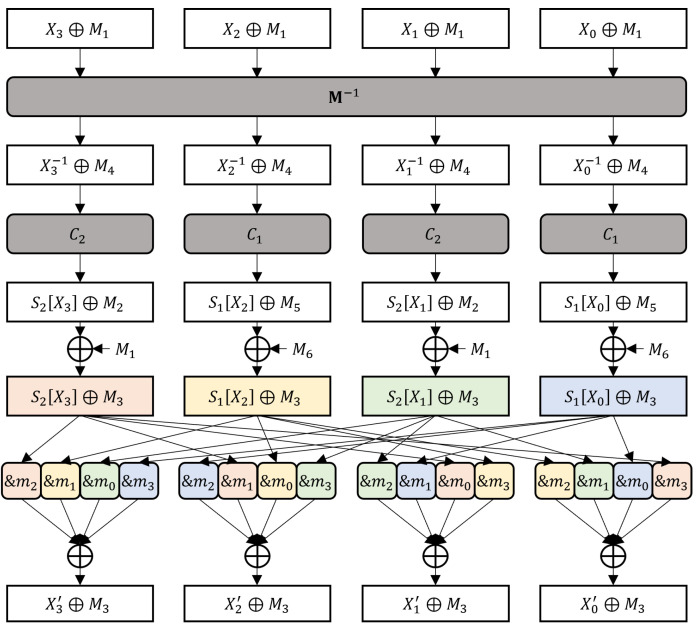
Masked function *G* of the existing masking scheme.

**Figure 3 sensors-24-05894-f003:**
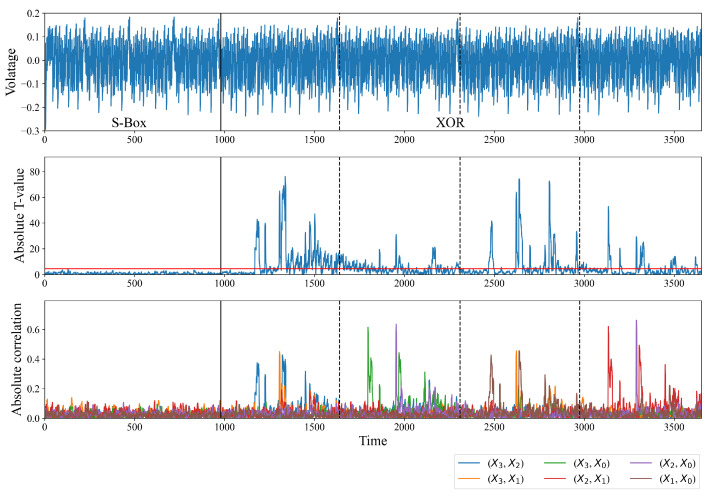
Point-wise T-values and absolute correlation coefficients of the proposed intermediate value over 10,000 traces. The XOR part is the output byte-wise AND and XOR calculations. (Top: power consumption, middle: absolute T-value, bottom: absolute correlation coefficient of the correct key.)

**Figure 4 sensors-24-05894-f004:**
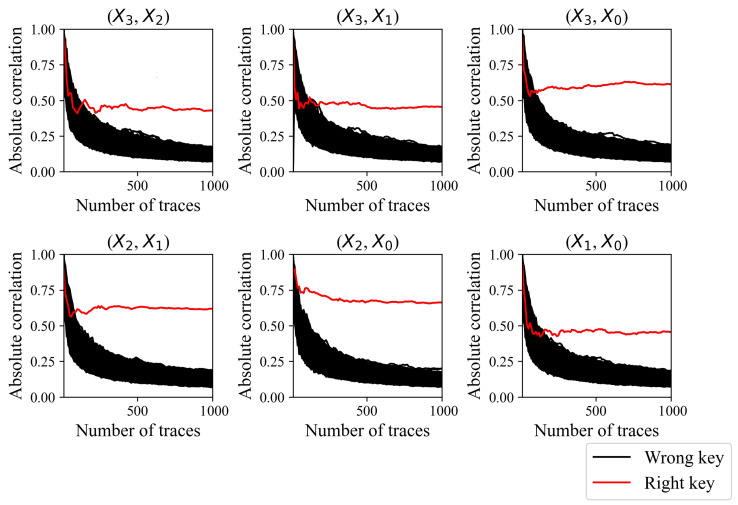
Absolute correlation coefficients of the correct key and incorrect keys according to the number of traces used.

**Figure 5 sensors-24-05894-f005:**
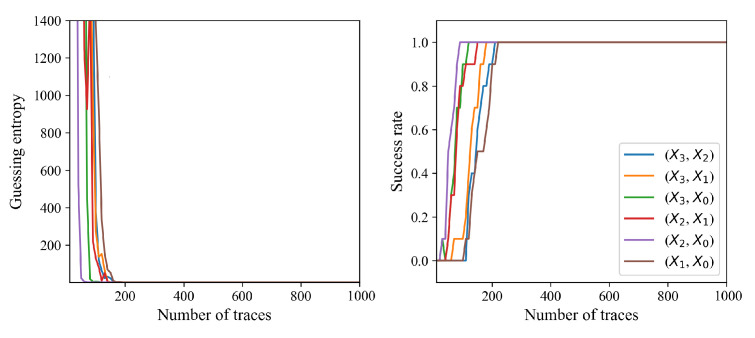
Guessed entropy and success rate derived from 10 CPAs.

**Figure 6 sensors-24-05894-f006:**
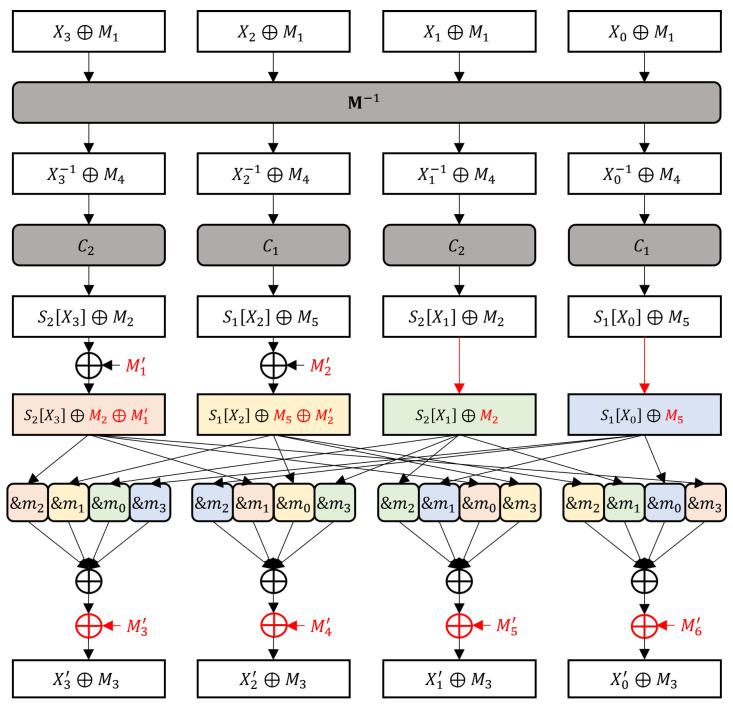
Masked function *G* of the proposed masking scheme.

**Figure 7 sensors-24-05894-f007:**
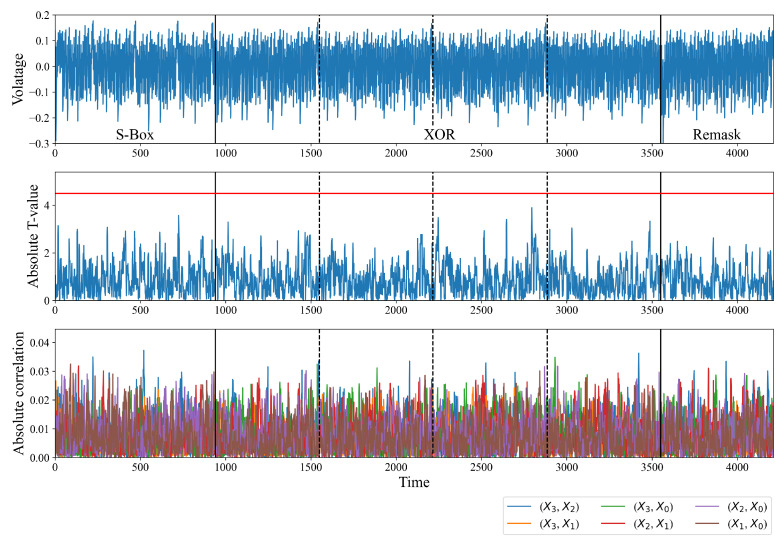
Point-wise T-values and absolute correlation coefficients of the proposed intermediate value over 10,000 traces. The XOR part is the output byte-wise AND and XOR calculations. The remask part changes the mask to $M_{3}$ after XORing the four S-Box outputs. (Top: power consumption, middle: absolute T-value, bottom: absolute correlation coefficient of the correct key.)

**Figure 8 sensors-24-05894-f008:**
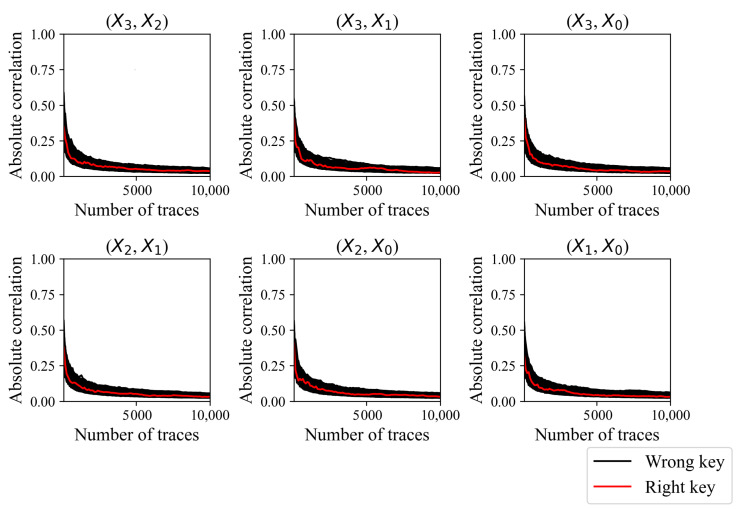
Absolute correlation coefficients of the correct key and incorrect keys according to the number of traces used.

**Figure 9 sensors-24-05894-f009:**
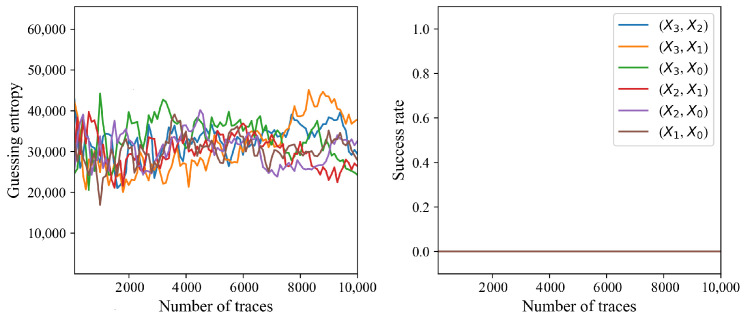
Guessed entropy and success rate derived from ten CPAs.

**Table 1 sensors-24-05894-t001:** Number of clock cycles for each SEED component.

	Existing Masking (Clock Cycle)	Proposed Masking (Clock Cycle)	Ratio
Masking generation	7174	7246	101.00%
Function G	136	160	117.65%
Entire encryption	153,821	154,973	100.75%

## Data Availability

The raw traces used in our demonstrations are available from the following link: https://1drv.ms/f/s!AjgtNjMfEUZWioNDyyzWaH0chGn3FQ?e=f0KP75 (accessed on 8 September 2024).

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
