# Peer review of "Tiny Security Hole: First-Order Vulnerability of Masked SEED and Its Countermeasure"

_sensors, 2024, doi:10.3390/s24185894_

Round 1
Reviewer 1 Report
Comments and Suggestions for Authors
To prevent the side-channel analysis, several masking methods of the ISO/IEC standard encryption algorithm SEED were proposed as the Korean financial IC card standard mandates. In this paper, the author proposed the first-order vulnerability of the masking method, where an adversary can perform side-channel analysis at the same complexity as the unprotected implementation. To fix the vulnerability, the authors revised the masking method at the costs of negligible additional overhead. The vulnerability and security are theoretically verified and experimentally demonstrated.
It is original and helpful, it can be acceoted after minor revision.
The motivation or purpose of this manuscript not clear enough.
The contributions should be described concisely.
Where are the limitations of your study? The limitations of a study allows the readers to understand better under which conditions the results should be interpreted.
Future work should be discussed.
The conclusion should be more informative.
Comments on the Quality of English LanguageThe expression is acceptalbe though some mistakes should be corrected.
Author Response
We sincerely appreciate the thoughtful feedback and the positive evaluation of our work. We have carefully considered the suggestions and have made the following revisions to the manuscript:
Comment 1: The motivation or purpose of this manuscript not clear enough.
Response 1: We clarified the motivation of the proposed method at the end of introduction (section 1) and beginning of subsection 3.1. We added the motivation "Most Boolean masking schemes struggle to reduce the overhead related to the substitution operation. Linear operations require a few additional XOR operations because they generally consist of linear operations. In contrast, the substitution operation requires a precomputed table related to the mask. Consequently, the security designer typically focuses on constructing a masking scheme related to the substitution operation; security evaluators also target the substitution operation output. Our motivation is that the intermediate value of the linear operation could be exploited to disclose secret information with much lower attack complexity. The security hole could occur while combining substitution layer outputs in linear operation." at the end of introduction. We revised the original statement "The motivation of the proposed method is that the operands of the last XOR operation of the function G are concealed with the same mask M3, and the AND operation before the XOR simply extracts bits of the operand. Therefore, the mask M3 is temporarily canceled during the last XOR." to "The motivation of the proposed method is that the operands of the last XOR operation of the function G are concealed with the same mask M3. As XORing two masked intermediate values induces canceling the mask, the adversary is able to exploit the XOR output as an intermediate value of the first-order CPA. Even though the operands are ANDed before XOR operation, a partial bit of mask is still canceled." in beginning of subsection 3.1
Comment 2: The contributions should be described concisely.
Response 2: In order to articulate the contributions of the manuscript, we have organized the contributions as follows: 1) Proposing the novel vulnerability of the SEED masking method via analyzing the structure of masking method. 2) Redesigning the masking method to fix the proposed vulnerability. 3) Demonstrating the vulnerability of existing masking method and security of proposed masking method by performing TVLA and CPA on real device. We have refined the description of our contributions for enhanced clarity.
Comment 3: Where are the limitations of your study? The limitations of a study allows the readers to understand better under which conditions the results should be interpreted.
Response 3: We clarified the limitation at the end of subsection 3.1:
"The limitation of the proposed method is that it may reveal only three of four bytes of the round key depending on the order of the XOR operation. Suppose that every four bytes of function G output is calculated by XORing four S-Box inputs in the same order. In this case, the round key associated with the last operand is unrelated to the proposed intermediate value, and the adversary reveals only three bytes of the round key; the round key related to the last operand is not revealed. On the other hand, if XOR calculates four operands in different orders, then the adversary reveals four bytes of round key."
Comment 4: Future work should be discussed.
We added the future work at the end of conclusion (section 5):
"The proposed security hole could also be present in other encryption algorithms. Although the AND operation is insufficient to prevent mask canceling, the security designer might mistakenly design the masking scheme to conceal the XOR operand with the same mask, like SEED. For instance, the block cipher ARIA also performs XOR on four S-Box outputs after the AND operation. Consequently, some masking methods of ARIA might also have the same security holes as SEED masking. Our future work is to investigate another masking scheme that has the proposed vulnerability and patch that."
Comment 5: The conclusion should be more informative.
We added the impact of the proposed vulnerability. We clarified that the security hole can be exploited to disclose the secret key with a same complexity as the unprotected implementation.
Reviewer 2 Report
Comments and Suggestions for Authors
The paper addresses the first-order vulnerability issue in SEED's masking implementation method against system side-channel analysis attacks. To mitigate this vulnerability, the authors have revised the masking method by incorporating additional XOR operations to mask intermediate values. The results demonstrate that the proposed masking method successfully prevents key disclosure even with 10,000 traces. While the paper is well-organized and supported by experiments, it lacks novelty as it primarily relies on commonly used security optimization methods.
The following comments may help to improve the quality of this paper.
1) The paper should analyze whether the modification to the ISO/IEC standard encryption algorithm has a negative impact on security, despite its improvement of the SEED’s ability to resist side-channel analysis attacks.
2) Please update of the related works and references, as many have been published for over 10 years.
Author Response
We sincerely appreciate the thoughtful feedback and the positive evaluation of our work. We have carefully considered the suggestions and have made the following revisions to the manuscript:
Comment 1: The paper should analyze whether the modification to the ISO/IEC standard encryption algorithm has a negative impact on security, despite its improvement of the SEED’s ability to resist side-channel analysis attacks.
Response 1: We would like to clarify that the masking countermeasure, including the proposed masking method, does not modify encryption input and output. This means that the algorithm encrypts the plaintext equivalent to the unprotected algorithm. As a result, the masked SEED has the equivalent security to the unmasked SEED.
Comment 2: Please update of the related works and references, as many have been published for over 10 years.
Response 2: We added four references at the end of introduction (section 1).
Won, Y.; Park, A.; Han, D. Novel Leakage Against Realistic Masking and Shuffling Countermeasures - Case Study on 348 PRINCE and SEED. In Proceedings of the Information Security and Cryptology - ICISC 2017 - 20th International Conference, 349 Seoul, South Korea, November 29 - December 1, 2017, Revised Selected Papers; Kim, H.; Kim, D., Eds. Springer, 2017, Vol. 350 10779, Lecture Notes in Computer Science, pp. 139–154. https://doi.org/10.1007/978-3-319-78556-1_8. 351
Messerges, T.S. Using Second-Order Power Analysis to Attack DPA Resistant Software. In Proceedings of the Cryptographic 352 Hardware and Embedded Systems - CHES 2000, Second International Workshop, Worcester, MA, USA, August 17-18, 353 2000, Proceedings; Koç, Ç.K.; Paar, C., Eds. Springer, 2000, Vol. 1965, Lecture Notes in Computer Science, pp. 238–251. 354 https://doi.org/10.1007/3-540-44499-8_19. 355
Herbst, C.; Oswald, E.; Mangard, S. An AES Smart Card Implementation Resistant to Power Analysis Attacks. In 356 Proceedings of the Applied Cryptography and Network Security, 4th International Conference, ACNS 2006, Singapore, 357 Version August 16, 2024 submitted to Sensors 14 of 14 June 6-9, 2006, Proceedings; Zhou, J.; Yung, M.; Bao, F., Eds., 2006, Vol. 3989, Lecture Notes in Computer Science, pp. 239–252. 358 https://doi.org/10.1007/11767480_16. 359.
Kim, J.H.; Sim, B.Y.; Han, D.G. SIV: Raise the Correlation of Second-Order Correlation Power Analysis to 1.00. Applied 364 Sciences 2020, 10. https://doi.org/10.3390/app10103394.
Reviewer 3 Report
Comments and Suggestions for Authors
- The manuscript presents a vulnerability discovered in masked SEED encryption algorithm. Given the state of urgency and severity of the security risk, the proposed security measures to fix the discovered first-order vulnerability causes negligible overhead, which is supported by the experimental evidence provided.
- The current method performs XOR operation between the intermediate values with the same mask, and this vulnerability allows an adversary to reveal the secret key with the same complexity as unmasked SEED, rendering the masking rudimentary.
- The proposed solution leverages the masking of first two bytes of the output of the s-boxes. It does not use an additional random mask, because generating a random mask substantially increases the computational requirements due to compute extensive operation like generating random numbers. However, the proposed method only requires 2 XOR operations per function. Given minor additional overhead in terms of memory and operation, the validity of the security of the proposed masking scheme is provided as well.
- The experimental evaluations like TVLA and CPA are performed to show both the vulnerabilities and the effectiveness of the proposed masking scheme. It shows that the proposed scheme successfully conceals the intermediate values that were at risk from the side channel cryptanalysis. Overall, the manuscript addresses the security risk from side channel attacks, while demonstrating the effectiveness of proposed solution through theoretical and experimental evidence.
Author Response
Comment: The manuscript presents a vulnerability discovered in masked SEED encryption algorithm. Given the state of urgency and severity of the security risk, the proposed security measures to fix the discovered first-order vulnerability causes negligible overhead, which is supported by the experimental evidence provided.
The current method performs XOR operation between the intermediate values with the same mask, and this vulnerability allows an adversary to reveal the secret key with the same complexity as unmasked SEED, rendering the masking rudimentary.
The proposed solution leverages the masking of first two bytes of the output of the s-boxes. It does not use an additional random mask, because generating a random mask substantially increases the computational requirements due to compute extensive operation like generating random numbers. However, the proposed method only requires 2 XOR operations per function. Given minor additional overhead in terms of memory and operation, the validity of the security of the proposed masking scheme is provided as well.
The experimental evaluations like TVLA and CPA are performed to show both the vulnerabilities and the effectiveness of the proposed masking scheme. It shows that the proposed scheme successfully conceals the intermediate values that were at risk from the side channel cryptanalysis. Overall, the manuscript addresses the security risk from side channel attacks, while demonstrating the effectiveness of proposed solution through theoretical and experimental evidence.
Response: I would like to sincerely appreciate your positive evaluation of my manuscript. As you mentioned, our manuscript aims to analyze and fix the vulnerability of the existing masking scheme. We are encouraged by your feedback and believe it will help us refine and enhance our work further. Thank you once again for your thoughtful review.
Reviewer 4 Report
Comments and Suggestions for Authors
The authors in this work demonstrated the vulnerability from a side channel attack perspective of the CEED algorithm. And they offered an amended version to rectify its vulnerability.
However their claim for minimun overhead is not demonstrated through experiments. Please add an experiment that proves the aforementioned claim and what is the expected cost for their amendment in the existing systems.
Author Response
Comment: The authors in this work demonstrated the vulnerability from a side channel attack perspective of the CEED algorithm. And they offered an amended version to rectify its vulnerability.
However their claim for minimun overhead is not demonstrated through experiments. Please add an experiment that proves the aforementioned claim and what is the expected cost for their amendment in the existing systems.
Response: We sincerely appreciate the thoughtful feedback and the positive evaluation of our work. We added number of clock cycles obtained from a real device at the end of subsection 4.2. Additionally, we have provided a comparison between the clock cycles of the existing masking method and those of the proposed masking method.
Round 2
Reviewer 2 Report
Comments and Suggestions for Authors
My concerns have been addressed.